# Research on the eLoran Differential Timing Method

**DOI:** 10.3390/s20226518

**Published:** 2020-11-14

**Authors:** Yun Li, Yu Hua, Baorong Yan, Wei Guo

**Affiliations:** 1National Time Service Center, Chinese Academy of Sciences, Xi’an 710600, China; hy@ntsc.ac.cn (Y.H.); yanbaorong@ntsc.ac.cn (B.Y.); guowei@ntsc.ac.cn (W.G.); 2Physics Department, University of the Chinese Academy of Sciences, Beijing 100049, China

**Keywords:** eLoran, difference, timing service, error, correlation

## Abstract

An enhanced long-range navigation (eLoran) system was selected as the backup of Global Navigation Satellite Systems (GNSS), and experts and scholars are committed to improving the accuracy of the eLoran system such that its accuracy is close to the GNSS system. A differential method called eLoran differential timing technology is applied to the eLoran system, which has been used in maritime applications of eLoran. In this study, an application of eLoran differential timing technology in a terrestrial medium is carried out. Based on the eLoran timing service error, the correlation of the timing service error is analyzed in theory quantitatively to obtain the range of the difference station in the ground. The results show that to satisfy the timing accuracy of 100 ns, the action range of eLoran difference station on the land needs to be less than 55 km. Therefore, the eLoran differential method is proposed, and in the difference station, the theoretical calculation is combined with the measurement of the signal delay to obtain the difference information, which is sent to the users to adjust the prediction delay and improve the eLoran timing precision. The experiment was carried out in the Guan Zhong Plain, and the timing error of the user decreased from 394.7287 ns (pre-difference) to 19.5890 ns (post-difference). The proposed method is found to effectively enhance the timing precision of the eLoran system within the scope of action.

## 1. Introduction

After delaying the instalment of enhanced long-range navigation (eLoran) stations for a certain period of time, the United States announced, on 4 December 2018, that eLoran stations will be used as a backup for Global Navigation Satellite Systems (GNSS). Moreover, simultaneously, a high-precision terrestrial time service system is being developed in China. The effective radiated power of eLoran is in the range of 100 to 1000 kW, and it is nearly impossible to jam over large areas with such high power levels. This is the reason why eLoran is used as an alternative to the GNSS system. Unfortunately, eLoran is less accurate than GNSS.

Several studies have been conducted to improve the precision of the eLoran system to match that of the GNSS system. Many experts and scholars focus on the prediction of signal propagation delay, especially the additional secondary factor (ASF) under special topographic conditions [1,2,3,4]. By using the ASF grid, the accuracy of the eLoran system can be improved [5,6,7,8]. To develop the ASF grid, surveys have been conducted for the ASF values. However, conducting investigation and measurement of ASF for the entire area is expensive. Moreover, ASF depends on the ground conductivity along the propagation path of the signal, and any change in conductivity due to weather and season will affect ASF. To supply the real-time variation of the ASF, the eLoran differential timing method has been proposed, especially for maritime applications, in which numerous experiment and data analysis are performed [5,6].

In this study, the eLoran differential timing service is analyzed theoretically and this method is mainly used in terrestrial media. The correlation of eLoran timing service error is analyzed in theory quantitatively, and the action range of difference station on the ground is calculated with an accuracy of 100 ns. Based on the correlation of the errors, the eLoran differential timing service is proposed. Then an experiment is carried out to test and verify the eLoran differential timing method by conducting a survey in the GuanZhong Plain of the Shanxi province.

## 2. eLoran Signal and Propagation Delay

### 2.1. eLoran Signal

eLoran is a low frequency (100 kHz) terrestrial navigation and timing system. The transmitters are organized into chains, with one master station and several secondary stations in each chain. The master station transmits nine pulses and secondary stations transmit eight pulses at a time. Figure 1 illustrates the shape of the signal, and the red dot represents standard zero-crossing (SZC), which is the point at which the signal is tracked by an eLoran receiver and it is used to calculate time-of-arrival.

### 2.2. eLoran Propagation Delay

The eLoran signal is transmitted radially from the transmitter to the receiver, and travels parallel to the surface of the earth. During this propagation, the signal does not travel at the speed of light, but it is slowed by the atmosphere and the surface of the earth. The time taken for the eLoran signal to reach the receiver is called propagation time ($T_{p}$), which is shown in the following equation
(1)$$T_{p}=\mathrm{PF}+\mathrm{ASF},$$
where if the signal propagates in the infinite air medium, the time taken to reach the receiving antenna from the transmitting antenna is called the primary factor (PF). Owing to the dielectric properties of the earth surface, the signal will travel relatively slowly as compared to that in the atmosphere. This delay is termed as an ASF.

The refractive index of the atmosphere $n_{s}$ implies that the speed of the signal is a fraction slower than the speed of light in vacuum. The PF is related to the distance *S*, and this is expressed as follows (Equation (2)) [9,10].
(2)$$\mathrm{PF}=\frac{\int n_{s}dS}{C}$$
where C is the speed of light in a vacuum and is equal to 0.299792458 km/μs, *S* is the distance of signal propagation, and $n_{s}$ is the atmospheric refraction index of the ground, which varies with temperature humidity and air pressure.

The earth surface medium (seawater and land) delay signal transmission, and the time taken for the eLoran signal in the procedure is ASF. With the decrease in the electrical conductivity of the surface, a high proportion of the signal will penetrate the ground and the wave-front will propagate more slowly. The calculation formula of ASF is shown in Equation (3) [9,10].
(3)$$\mathrm{ASF}=\frac{10^{6}}{\omega }\arg W(f,d,\sigma ,\varepsilon ),$$
where ω is the angular velocity in rad/s, W is the signal attenuation function, which is related to signal frequency f, the distance of signal propagation d, the electrical conductivity σ and dielectric constant ε of signal propagation path, and argW is the phase of the W in rad.

## 3. Principle of eLoran Timing Service and Error Analysis

### 3.1. Principle of eLoran Timing Service

The transmitter broadcasts the timing signal, which propagated in the channel and arrived at the receiver. In the receiver, the signal is processed and demodulated to obtain the one-pulse-per-second (1PPS), and then the receiver local 1PPS signal is synchronized with the received 1PPS. Specifically, the group time pulse (GTP) signal received by the eLoran receiver is compared with the local 1PPS of the receiver, and then the local 1PPS signal is adjusted according to the offset Δτ [11,12,13]. This principle is illustrated in Figure 2.

After adjusting the local 1PPS pulse of the receiver to be ahead of 1PPS (UTC), the local 1PPS of the receiver is used as the opening signal of the counter, and the GTP signal demodulated by the receiver is used as the closing signal. Subsequently, the phase difference N is measured between these two pulses as shown below (Equation (4)).
(4)$$N=\Delta \tau +\left(t_{0}+t_{p}+t_{r}\right)$$
where $t_{0}$ is the transmission delay of the system obtained from the demodulated message, $t_{g}$ is propagation delay calculated with the position of transmitter and receiver, and $t_{r}$ is the receiver delay, which is calibrated in advanced. Then, the offset of local 1PPS signal Δτ is obtained using Equation (5).
(5)$$\Delta \tau =N-\left(t_{0}+t_{p}+t_{r}\right),$$

The errors of N, $t_{0},t_{p},t_{r}$ are transferred to Δτ. The time difference N is measured by the inner counter of the receiver, and an error is introduced in the procedure. $t_{0}$ can be calibrated accurately, and the error after calibration is approximately 30 ns. $t_{p}$ is calculated using Equations (1)–(3), and the deviation from the true value is approximately 500 ns. $t_{r}$ is measured with an error less than 20 ns. The majority of the timing error is due to the error in $t_{0},t_{p},t_{r}$ which will be addressed in a future study.

### 3.2. Error Analysis of eLoran Timing Service

From the source to the destination, a signal from a source passes through three systems to reach the destination. These three systems are the transmitter, propagation channel, and receiver. During this process, the entire propagation delay is composed of transmission delay, signal propagation delay and signal receiving delay, which are shown in Figure 3. The signal transmission delay error, signal propagation delay error, and signal receiving delay error make up the eLoran timing error.

#### 3.2.1. eLoran Transmission Delay Error

The eLoran signal transmission error includes the synchronization error between the local time in a broadcast station and UTC (NTSC), the transmitter channel delay error and the antenna phase center error.

(1)Synchronization error between the local time in the station and UTC (NTSC)

The broadcasting system of eLoran is equipped with an independent atomic clock which is used as the time reference and keeps synchronous with UTC (NTSC) through a certain communication link. The synchronization error is generally less than 5 ns.

(2)Transmitter channel delay error

The transmitter channel delay error comprises signal coding delay error and delay error in the signal modulation, which can be controlled to be within 30 ns.

(3)Error of antenna phase center

In theory, the phase center of the antenna should be consistent with its geometric center, but the change in the signal carrier frequency can offset the phase center, which leads to the uncertainty of the antenna phase center. In the 100 kHz carrier frequency band, the phase center fluctuation of the antenna is less than 5 ns.

The estimated eLoran transmission delay error is calculated using the following equation $\tau_{0}=\sqrt{5^{2}+30^{2}+5^{2}}=30.4138\;\mathrm{ns}$.

#### 3.2.2. eLoran Propagation Delay Error

The true value of signal propagation delay is shown in Equation (6). However, it is difficult to obtain the true value of the $n_{s}$, σ and ε. In eLoran timing service, the calculation is generally used to obtain the signal propagation delay. In the calculation, the value of $n_{s}^{\prime }$ is 1.000315 in the international standard atmosphere, and the quantization value $n_{s}^{\prime },\sigma^{\prime }$ and $\varepsilon^{\prime }$ replace the true $n_{s}$, σ and ε. This is the main reason for the delay error of signal transmission. For Equation (7), the deviation from Equation (6) is the propagation delay error shown in Equation (8), where ΔPF represents the error of PF and ΔASF is the error of ASF. At present, the error of the eLoran signal propagation delay is better than 500 ns.
(6)$$T=\mathrm{PF}+\mathrm{ASF}=\frac{\int n_{s}dS}{C}+\frac{10^{6}}{\omega }\arg W\;(f,d,\sigma ,\varepsilon )$$
(7)$$T^{\prime }={\mathrm{PF}}_{p}+{\mathrm{ASF}}_{p}=\frac{{Sn^{\prime }}_{s}}{C}+\frac{10^{6}}{\omega }\arg W\;(f,d,\sigma^{\prime },\varepsilon^{\prime })$$
$$\Delta T=T-T^{\prime }$$
$$=\left(\mathrm{PF}+\mathrm{ASF}\right)-\left({\mathrm{PF}}_{p}+{\mathrm{ASF}}_{p}\right)$$
$$=\left(\mathrm{PF}-{\mathrm{PF}}_{p}\right)\left(\mathrm{ASF}-{\mathrm{ASF}}_{p}\right)$$
=ΔPF+ΔASF
(8)$$=\frac{\int n_{s}dS}{C}-\frac{Sn_{s}}{C}+\frac{10^{6}}{\omega }\left[\arg W(f,d,\sigma ,\varepsilon )-\frac{10^{6}}{\omega }\arg W(f,d,\sigma ,\varepsilon )\right].$$

#### 3.2.3. eLoran Receiving Delay Error

The receiving delay is the time from the moment when the signal propagates to the receiving antenna to the moment when the receiver outputs 1PPS [14,15,16]. The receiver delay error includes the thermal noise of the receiver and the measurement error introduced during receiver delay measurement [17,18]. After calibrating the receiver’s delay, the receiving delay error can be controlled to be within 20 ns. All the source of errors in eLoran timing is listed in Table 1.

## 4. Correlation Analysis of eLoran Timing Error

The differential method used in eLoran timing service is similar to the GPS local difference technology. According to the GPS differential technology, the error correlations between the difference station and the user in an area lay the foundation of the difference technology. The timing error correlation between the difference station and the user in an area is analyzed sequentially. In Figure 4, A is the difference station and B represents users. In the two paths OA and OB, weather, geology, and geography are similar, which is the precondition of the difference method.

### 4.1. Correlation of eLoran Transmitting Delay Error

Transmission delay error is one of the key errors that need to be corrected. Difference stations and nearby users receive signals from the same transmitting station. Regardless of the location of the difference station and the user, the influence of the transmission error on the difference station and the user is the same. If the error is 10 ns, it will induce 10 ns timing error and 3 m pseudo-range error to users and the difference station [19].

### 4.2. Correlation of eLoran Propagation Delay Error

The propagation delay of eLoran signal comprises PF and ASF. Although these two factors are indistinguishable in propagation delay measurement, their physical properties vary considerably. Therefore, the error correlation of PF and the error correlation of ASF is discussed separately.

#### 4.2.1. The error Correlation of PF

$e_{M}={\mathrm{PF}}_{M}-{{\mathrm{PF}}^{\prime }}_{M}$ is the PF error of difference station and $e_{U}={\mathrm{PF}}_{U}-{{\mathrm{PF}}^{\prime }}_{U}$ is the PF error of the users, and the difference between the two is analyzed using Equation (9). First, assuming that the difference between the distance of two signal propagation path is Δd, and the amount of $\Delta n_{s}$ is not more than the change in $n_{s}$ within one year in the area, such that $\Delta n_{s}$ is not greater than 0.000060 [20,21], then the difference between the primary delay error caused by $n_{s}$ is expressed as follows.
$$\left|e_{U}-e_{M}\right|=\left|\left({\mathrm{PF}}_{U}-{{\mathrm{PF}}^{\prime }}_{U}\right)-\left({\mathrm{PF}}_{M}-{{\mathrm{PF}}^{\prime }}_{M}\right)\right|$$
$$=\left|\left(\frac{d_{U}\times n_{s}}{C}-\frac{d_{U}\times {n^{\prime }}_{s}}{C}\right)-\left(\frac{d_{M}\times n_{s}}{C}-\frac{d_{M}\times {n^{\prime }}_{s}}{C}\right)\right|$$
$$=\left|\frac{d_{U}\times \Delta n_{s}}{C}-\frac{d_{M}\times \Delta n_{s}}{C}\right|$$
$$=\left|d_{U}-d_{M}\right|\times \frac{\Delta n_{s}}{C}$$
(9)$$\leq \Delta d\times \frac{0.000060}{0.299792458}.$$

#### 4.2.2. Error Correlation of ASF

$e_{M}={\mathrm{ASF}}_{M}-{{\mathrm{ASF}}^{\prime }}_{M}$ is the ASF error of difference station and $e_{U}={\mathrm{ASF}}_{U}-{{\mathrm{ASF}}^{\prime }}_{U}$ is the ASF error of the user, and the difference between the two is analyzed using Equation (10). ${\mathrm{ASF}}_{M}$ is the ASF with true σ and ε for the difference station, and the ${{\mathrm{ASF}}^{\prime }}_{M}$ is the ASF with the quantization value $\sigma^{\prime }$ and $\varepsilon^{\prime }$ for the difference station. ${\mathrm{ASF}}_{U}$ is the ASF with the true σ and ε, and the ${{\mathrm{ASF}}^{\prime }}_{U}$ is ASF with the quantization value $\sigma^{\prime }$ and $\varepsilon^{\prime }$ for the user.
$$\left|e_{U}-e_{M}\right|=\left|\left({\mathrm{ASF}}_{U}-{{\mathrm{ASF}}^{\prime }}_{U}\right)-\left({\mathrm{ASF}}_{M}-{{\mathrm{ASF}}^{\prime }}_{M}\right)\right|$$
$$=\left|\left({\mathrm{ASF}}_{U}-{\mathrm{ASF}}_{M}\right)-\left({{\mathrm{ASF}}^{\prime }}_{U}-{{\mathrm{ASF}}^{\prime }}_{M}\right)\right|$$
$$=\left|\left[F\left(d_{U},\sigma ,\varepsilon \right)-F\left(d_{M},\sigma ,\varepsilon \right)\right]-\left[F\left(d_{U},\sigma^{\prime },\varepsilon^{\prime }\right)-F\left(d_{M},\sigma^{\prime },\varepsilon^{\prime }\right)\right]\right|$$
(10)$$=\left|\Delta \mathrm{ASF}-\Delta {\mathrm{ASF}}^{\prime }\right|.$$

According to radio wave propagation theory, the calculation of ASF is quite complex, as shown in Equation (3). However, if Equation (3) is used, it is very inconvenient to study the error correlation of ASF in Equation (10). The relationship between ASF and distance *d* can be simplified by using another algorithm for convenience engineering calculation, but the residual error must meet the conditions of the military standard long-wave ground wave propagation [5]. When the signal propagation distance is greater than 100 km, the ASF and distance *d* are almost linear, and the relationship between the SF and distance is fitted with a linear polynomial ASF=a×d+b [22].

First, using Equation (3), the ASF is calculated as 0<d≤2000 and the relation between ASF and *d* is obtained on a typical propagation path. Second, the relationship between ASF and distance *d* is fitted by the least square algorithm using the calculated data. Finally, if the fitting residual error meets the conditions of the military standard, the fitting is termed reasonable. The simulation result is listed in Table 1, and the limit error set by the military standard is presented in the last column. The fitting on five types of paths are sufficient to satisfy the condition, but the fitting residual error of seawater and average land exceed the military standard, in Table 2.

In Figure 5, the fitting curve of the sea surface is shown in the top and that of the average land is on the bottom. The red line represents the result of the calculation and the blue line represents the fitting result. The maximum residual error appears near 100 km. With the increase in distance, the fitting results also improve. The distance between the difference station and the eLoran broadcast station is generally greater than 100 km. Therefore, for the sea surface and the average land path, it is also reasonable to fit the relationship between ASF and d with a polynomial SF=a×d+b, when the distance is more than 100 km.

For the typical signal propagation path, the value of the electrical conductivity has a certain range. For example, in the third column of Table 3, it can be seen that the electrical conductivity of the seawater is between 3 and 7, which are the minimum and the maximum values, respectively. When the maximum value of σ is considered, the corresponding fitting parameters are a and b and when the minimum value of σ is considered, the corresponding fitting parameters are $a^{\prime }$ and $b^{\prime }$. The variation in ASF due to uncertainty in σ uncertainty from seasonal and weather effects is smaller than the variation in ASF brought from maximum to the minimum of σ.

Based on the simplified relationship between ASF and distance d, the ASF error comparison between user and differential station is calculated using Equation (11). Including the PF and the ASF, the entire propagation delay error of the user and the differential station is compared using Equation (12).
$$\left|e_{U}-e_{M}\right|=\left|\Delta \mathrm{ASF}-\Delta {\mathrm{ASF}}^{\prime }\right|$$
$$=\left|a\left(d_{U}-d_{M}\right)-a^{\prime }\left(d_{U}-d_{M}\right)\right|$$
$$=\left|\left(a-a^{\prime }\right)\left(d_{U}-d_{M}\right)\right|$$
(11)$$=\left|\left(a-a^{\prime }\right)\times \Delta d\right|$$
(12)$$\left|e_{U}-e_{M}\right|=\Delta d\times \frac{0.000060}{0.299792458}+\left|\Delta d\times \left(a-a^{\prime }\right)\right|.$$

### 4.3. Correlation of Receiving Delay Error

The delay error of the receiver is primarily the thermal noise of the receiver equipment, which has no correlation [19,23], but we can reduce the error by measuring the relative delay of the receivers [24]. The residual error is less than 5 ns after relative delay measurement and calibration.

### 4.4. Correlation of the Entire Timing Error

After the correction by the difference station, the residual error for the user is shown in Table 4. How much is the distance from the user to the difference station, which can meet the timing accuracy of 100 ns. Using Equation (13), the distance is calculated and is listed in Table 5. On dryer ground path, the effective range is approximately 55 km, which is the minimum range. To achieve 100 ns precision, the effective range of the difference station on the land should be less than 55 km, in the condition of the distance greater than 100 km from the difference station to the broadcasting station.

$$\left|e_{U}-e_{M}\right|\leq D\times \left(\frac{0.000060}{0.299792458}+\left|a-a^{\prime }\right|\right)+0.005\leq 0.1$$

(13)$$\Rightarrow D\leq \frac{0.095}{\frac{0.000060}{0.299792458}+abs\left(\Delta a\right)}$$

## 5. The eLoran Difference Timing Method

Within 55 km around the difference station, there is a strong correlation with the timing error, and the offset between the difference station and the user timing error is not more than 100 ns. Therefore, we can package the uncertainty in the timing error of the difference station into differential correction and send them to the surrounding users to correct the propagation delay of the users and to improve the timing accuracy. For the difference station, the differential message is calculated by combining the measurement and calculation of signal propagation delay, and the differential correction model is generated and sent to the user. For users, the differential correction is calculated to modify the prediction value of propagation and improve the accuracy of time service. The calculation and prediction of the signal propagation delay are shown using Equation (7).

In a region, a site is selected as the difference station for surrounding users. In the difference station, it is important to measure the signal propagation delay to calculate the difference message. Measurement devices, such as eLoran monitoring receiver, time interval counter, data acquisition equipment, and UTC (NTSC) is used. Without UTC (NTSC), the local time-synchronized to the UTC (NTSC) is also enough.

For the difference station, first, the accurate position coordinate of the monitoring receiver antenna is calibrated accurately to calculate the distance from the transmitter to the receiver. Second, the signal propagation delay $T_{p}$ is calculated using Equation (7). Then, the monitoring receiver receives the eLoran signal and demodulate the 1PPS, which is compared with the UTC (NTSC) through the time interval counter to obtain the time difference *M*. Finally, the time difference M is collected to calculate the $M-T_{p}$ to form a differential model and is transmitted to the surrounding users in real-time.

For the user, with the precise position of the receiver antenna, the circle distance from the user to the broadcast station is calculated. The signal propagation delay ${T^{\prime }}_{p}$ is calculated according to Equation (7), then the correction is calculated according to the received difference information to obtain more accurate signal delay and synchronize the 1PPS to UTC (NTSC). The principle of eLoran differential time service is shown in Figure 6.

## 6. Verification of eLoran Differential Timing Method

The experimental research is carried out to verify the result of the difference technology. In the experiment, a survey is carried out in the difference station and the users, but with a different purpose. The measurement in difference station is to calculate the difference in correction mode, and the test in user site is to verify the difference technology effect. The offset between the difference result and the measured value of delay is used to evaluate the effect of the timing differential method. The schematic of the test equipment is shown in Figure 7. The BD receiver is used as the standard time and frequency source, which have been synchronized to the UTC (NTSC) in advance. The time interval counter is used to measure the signal delay, which is collected in the data acquisition system.

The measurement is carried out in GuanZhong Plain in Shanxi Province. The area has wet ground, with conductivity *σ* = 0.01 and dielectric constant *ε* = 30. The basic information of test points is shown in Table 6. Taking Wu Gong (WG) as the difference station and Mei Xian (MX) as the user, the distance between WG and MX is 44.8930 km. The distribution of test points is shown in Figure 8. Pu Cheng (PCH) is the eLoran broadcast station.

The measurement data in WG and MX is shown in Figure 9. From the changing trend of the curve, the variation of propagation delay in the MX is coincident with the WG over time. In WG, the measurement data and the theoretical calculation results are combined to compute the difference information. In the calculation and forecasting of the difference model, the least square principle is used for the difference station. The sliding window is 600 and the forecast time is 300 s, with 1 as the fitting order. In other words, the data in 10 min forecast for 5 min.

As shown in Figure 10, the standard error is reduced from 32.7356 ns pre-correction to 18.9630 ns post-correction. The average of all the errors in 394.7287 ns pre-difference, and the average of the error is 19.5890 ns after correction. The accuracy is improved by an order of magnitude. It is proved that the timing service accuracy of eLoran is improved by the difference method, and is within the range of the difference station.

## 7. Conclusions

First, eLoran signal and the propagation delay on the path was introduced, and then the eLoran timing error was analyzed. Second, the correlation of eLoran timing error was analyzed quantitatively in theory based on eLoran timing error and it was found that the action range of the difference station on land should not be more than 55 km to achieve 100 ns accuracy. Finally, based on the correlation between user and difference in timing error, an eLoran differential timing technology was proposed. In the difference station, the theoretical calculation method is combined with the measurement method of the signal delay to obtain the offset which is the common error for the difference station and the users. The offset data were then processed to obtain the difference information which is sent to the users so as to adjust the propagation delay and improve the timing precision.

To validate the correctness and validity of the proposed method, experimental verification was carried out in Guan. Zhong Plain. In the experiment, the measurement for the users and the difference station are carried out synchronously to verify the effect of differential technology. The timing precision of the users is improved from 394.7287 to 19.5890 ns, which shows that this method can significantly improve the timing accuracy of the system. Although the range of the difference station in the land is smaller than that of the maritime applications, it can effectively enhance the timing precision of the eLoran system within the action scope.

## Figures and Tables

**Figure 1 sensors-20-06518-f001:**
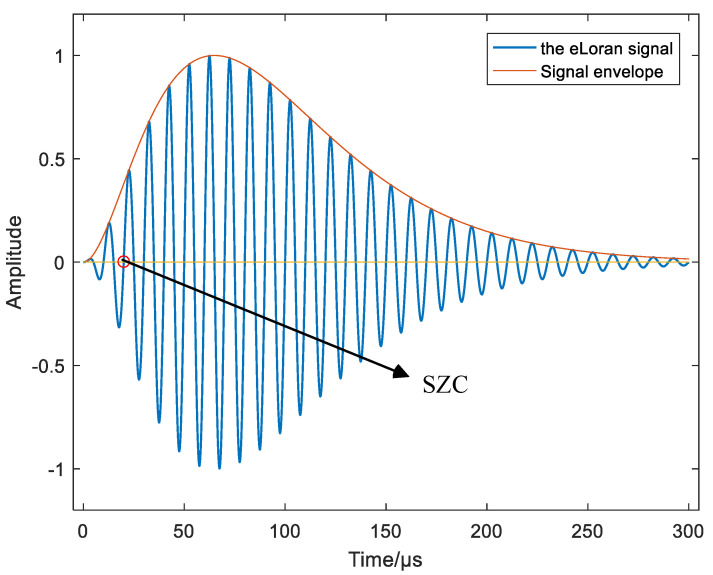
The enhanced long-range (eLoran) pulse shape. The red dot represents standard zero-crossing (SZC), which is the point at which the signal is tracked by an eLoran receiver and it is used to calculate time-of-arrival.

**Figure 2 sensors-20-06518-f002:**
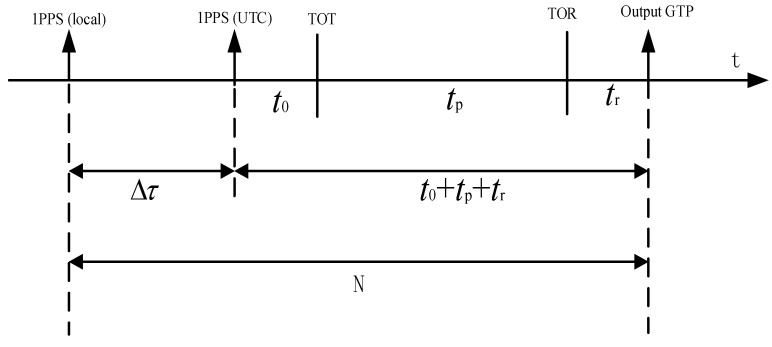
Principle of eLoran autonomous timing.

**Figure 3 sensors-20-06518-f003:**
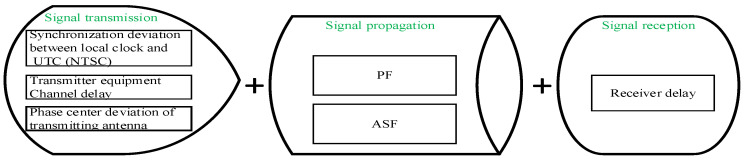
The entire propagation delay is composed of transmission delay, signal propagation delay and signal receiving delay. The signal propagation consists of the primary factor (PF) and the additional secondary factor (ASF).

**Figure 4 sensors-20-06518-f004:**
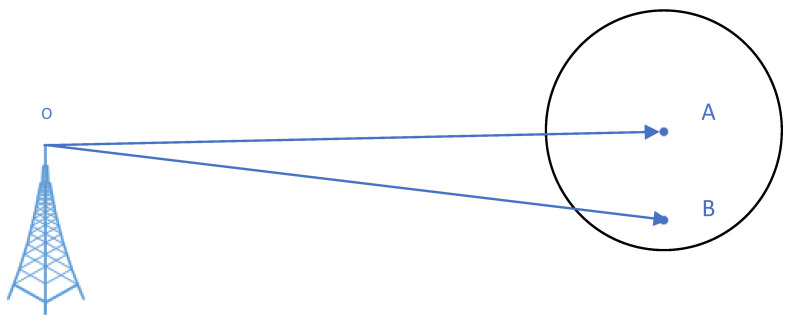
Diagram of difference station and surrounding users.

**Figure 5 sensors-20-06518-f005:**
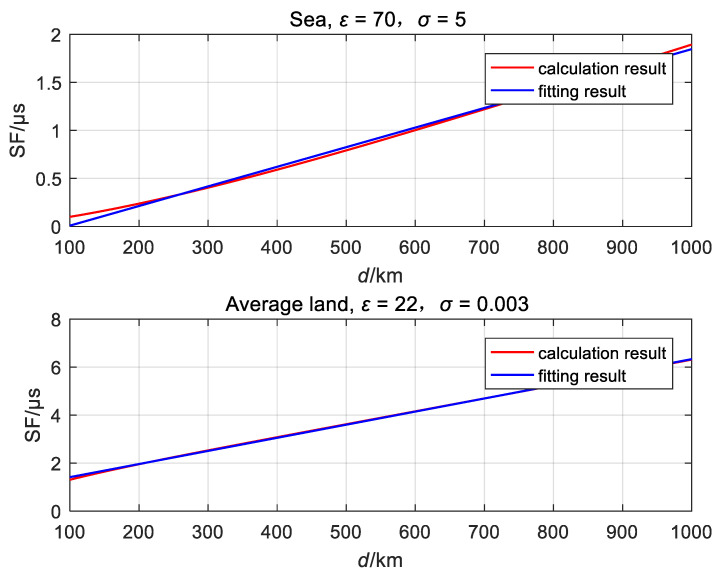
Fitting curve of seawater and average land.

**Figure 6 sensors-20-06518-f006:**
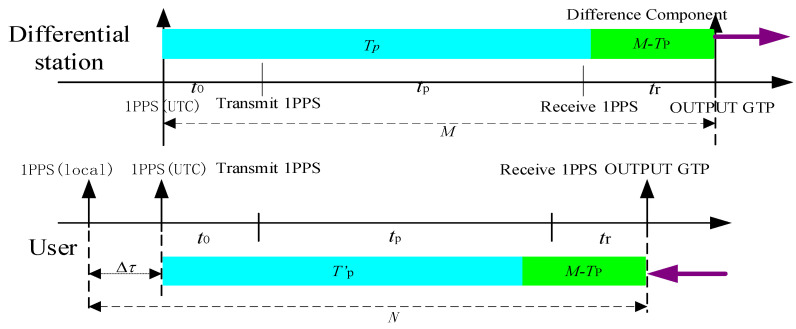
Principle of differential timing.

**Figure 7 sensors-20-06518-f007:**
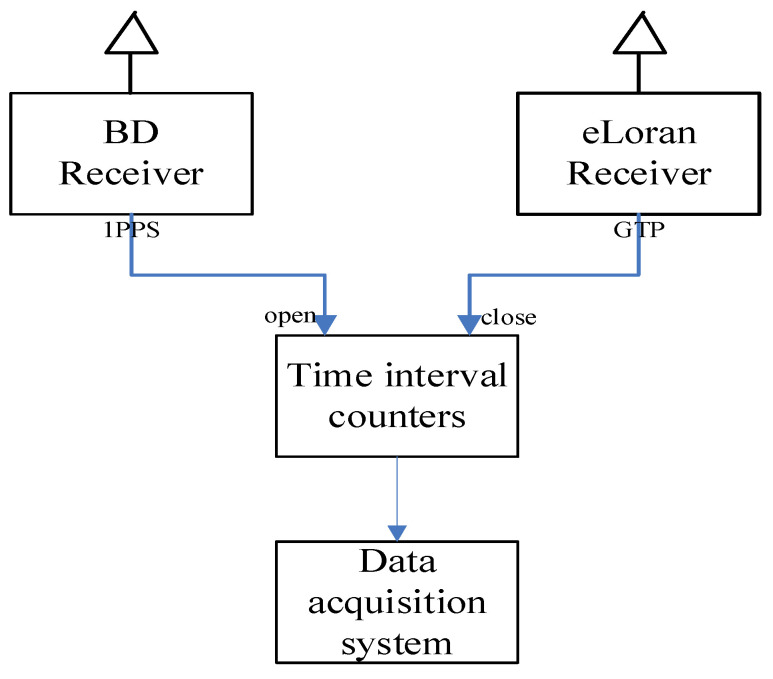
Measurement equipment connection.

**Figure 8 sensors-20-06518-f008:**
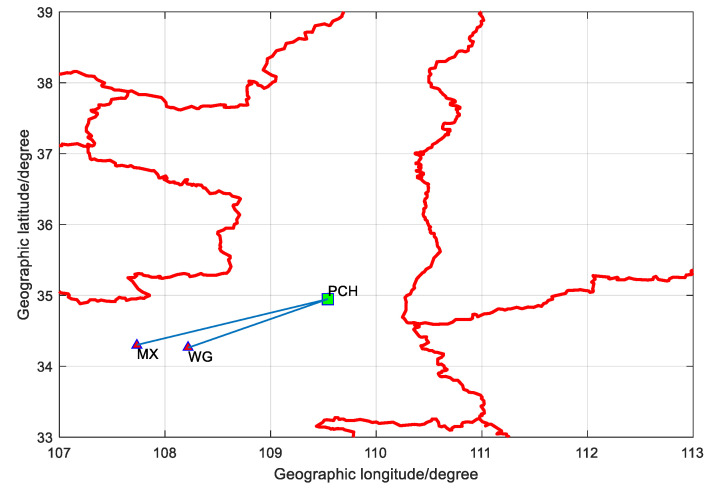
Distribution of test point.

**Figure 9 sensors-20-06518-f009:**
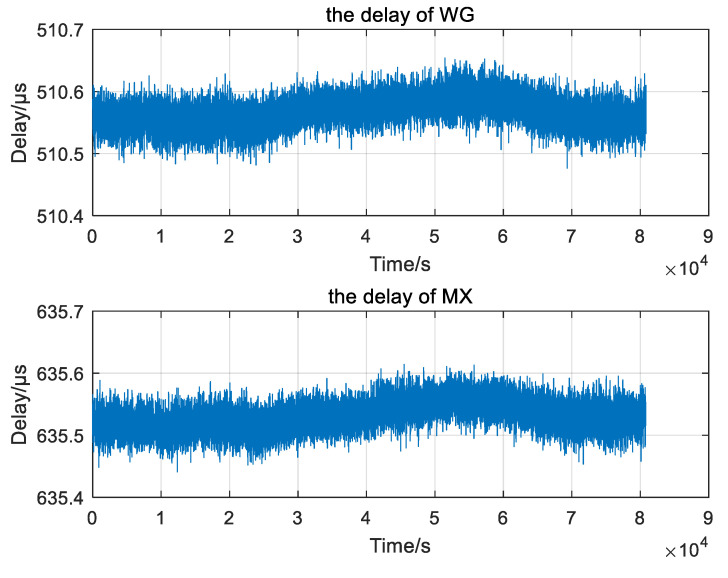
Delay measurement in Wu. Gong (WG) and Mei. Xian (MX).

**Figure 10 sensors-20-06518-f010:**
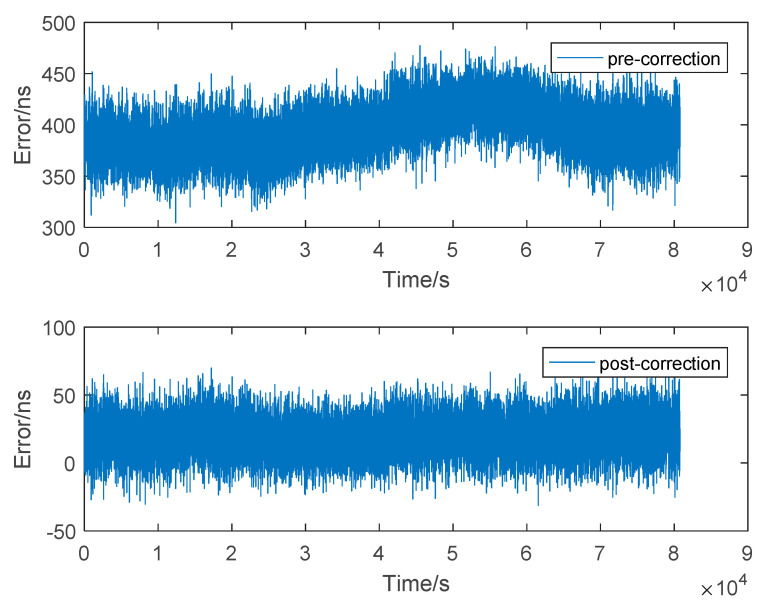
Effect of difference method in MX.

**Table 1 sensors-20-06518-t001:** Source of errors in eLoran timing.

Error Source	Magnitude/ns	Notes
transmission delay error	$\sqrt{5^{2}+30^{2}+5^{2}}$	Synchronization error between eLoran time and UTC(NTSC), transmitter channel delay error and error of antenna phase center
propagation delay error	500	The deviation between the calculation value and the true
receiving delay error	20	The delay error of eLoran receiver

**Table 2 sensors-20-06518-t002:** The fitting analysis of secondary delay (100 km < *d*).

Ground Type	Mean (μs)	Standard (μs)	Maximum (μs)	Maximum in Military Standard (μs)
Seawater	−4.9830 × 10^−16^	0.0304	0.0915	0.02
Land with good conduction	−1.2401 × 10^−15^	0.0156	0.0334	0.05
Wet ground	−9.9587 × 10^−16^	0.0061	0.0991	0.1
Average Land	−1.3052 × 10^−15^	0.0237	0.0251	0.2
Dry land	−2.5275 × 10^−15^	0.0722	0.0763	0.2
Dryer Ground	−3.3674 × 10^−15^	0.0900	0.1065	0.2
Driest Ground	−3.1111 × 10^−15^	0.0659	0.1257	0.2

**Table 3 sensors-20-06518-t003:** Additional secondary factor (ASF) fitting coefficient of the typical path.

Ground Type	ε	σ	Equivalent Earth Radius Coefficient	Fitting Coefficients			
Seawater	70	7	1.14	*a*	0.0020	*b*	−0.1996
		3		*a*′	0.0021	*b*′	−0.1878
Land with good conduction	40	0.055	1.13	*a*	0.0028	*b*	0.0292
		0.017		*a*′	0.0035	*b*′	0.2318
Wet ground	30	0.017	1.11	*a*	0.0035	*b*	0.2284
		0.0055		*a*′	0.0046	*b*′	0.5795
Average Land	22	0.0055	1.08	*a*	0.0047	*b*	0.5730
		0.0017		*a*′	0.0062	*b*′	1.2559
Dry land	15	0.0017	1.06	*a*	0.0063	*b*	1.2535
		0.00055		*a*′	0.0067	*b*′	2.4676
Dryer Ground	7	0.00055	1.05	*a*	0.0070	*b*	2.5488
		0.00017		*a*′	0.0055	*b*′	3.6182
Driest Ground	3	0.00017	1.06	*a*	0.0055	*b*	3.8882
		0.000055		*a*′	0.0049	*b*′	3.732

**Table 4 sensors-20-06518-t004:** Residual error after correction.

Error Item	Residual Error Post-Difference/μs
The error transmitting delay	0
The error of propagation delay	$D\times \frac{\Delta n_{s}}{C}+\left|\left(a-a^{\prime }\right)\times D\right|$
The error of receiving delay	0.005
sum	$D\times \frac{\Delta n_{s}}{C}+\left|\left(a-a^{\prime }\right)\times D\right|+0.005$

**Table 6 sensors-20-06518-t006:** The information of the measurement point.

Measure Point	Longitude	Latitude	The Distance/km	PF/μs	SF/μs	TOA
Wu. Gong	108.2200	34.2618	143.2942	478.1286	0.9422	479.0708
Mei. Xian	107.7348	34.3014	180.7000	602.9402	1.0964	604.0366

**Table 5 sensors-20-06518-t005:** Scope of difference station in a typical path.

Ground Type	abs(Δa)	The Range of Difference Station/km
Seawater	0.0001	316.6667
Ground with good conduction	0.0007	105.5556
Wet ground	0.0011	73.0769
Average Ground	0.0015	55.8824
Dry Ground	0.0004	158.3333
Dryer Ground	0.0015	55.8824
Driest Ground	0.0006	118.7500

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
