# Peer review of "Research on the eLoran Differential Timing Method"

_sensors, 2020, doi:10.3390/s20226518_

Round 1
Reviewer 1 Report
General remarks:
- Investigations on eLoran and the possibilities of increasing performance by means of differential correction approaches are not new. The decision of the USA to consider eLoran again as a backup option to GNSS has resulted in the fact that eLoran and differential correction approaches becoming the subject of investigations and potential further developments again. The paper fits into this context. It should also be mentioned that there are different views on eLoran as a GNSS backup system worldwide, including the maritime community. This should be corrected in the abstract and in the introduction of the paper.
- Spelling and grammar should be checked throughout the paper. Attention should also be paid to the formation of complete sentences.
Special remarks:
- Chapter 2 describes the propagation errors of eLoran signals, since only these can be corrected with differential approaches. The description is partly insufficient, partly redundant. Inadequate, because e.g. in the formula a strict separation of SF and ASF was made, which does not exist e.g. in coastal regions in such a way and has also effect on the effectiveness of differential approaches. Redundant, because e.g. the significant differences between equation 4 and 5, if there are any and if they are justified, are neither recognizable nor described. The section should be completely revised.
- Section 3.1 describes the error analysis of eLoran timing service. Here it would be helpful if measured values, error sizes and output values could be clearly identified and characterized. This would help to establish a better reference to the following chapters.
- Section 3.2 deals with the error analysis of eLoran timing service and only now classifies the propagation error into the total error budget. This should perhaps be done in the paper rather (section 2). The error considerations are on the one hand considering error limits (transmitter, receiver) and on the other hand estimating errors (propagation). This should also be reflected in the choice of words. Equation 10 is not error-free and must be checked. Also the question arises whether e.g. formula 10 must necessarily reflect all display options (The question also applies to Formula 11, 12 and 13).
- The remarks made in chapter 4 use e as error indication. In chapter 2, e was introduced as dielectric constant. This is not helpful for readability. In table 1 and 2 the source of the data deliveries should be recognizable: which basic data were obtained from where, what was calculated and which are in the focus of further considerations. The relation of table 2 to figure 3 should be clearly recognizable. Equation 15 can be derived unconditionally from equation 14. Therefore the explanations in line 205-208 are not comprehensible. Sections 4.3 and 4.4 have the same title, whereby the content is related to different questions. This should be resolved. Equation 17 is inconsistent. Title and content of Table 5 should be proved and set in relation to equation 17.
- Chapter 5 has to set in relation to chapter 4.
- The illustration and discussion of study results in chapter 6 and 7 should be improved, e.g. by comparison of this results with similar results and by elaborating the importance of the gathered results.
Author Response
Please see the attchment.

Reviewer 2 Report
My opinion is that the language has to be extensievely improved and the paper should be resubmitted after that is acheived.
Reviewer 3 Report
Dear Authors,
the letter entitled: Research on eLoran Differential Timing Method presents the application research of differential timing technology of terrestrial medium, start from the eLoran timing service error, quantitative analysis of the correlation of the timing service error, and give a reference value for the range of the difference station in the ground, then the eLoran differential technology is given and the method is verified by experiments. It is important paper because the eLoran system has been selected as the primary backup system of GNSS system.
Almost all chapters (abstract, introduction to the research topic, propagation delay of eLoran signal, the principle and error analysis of eLoran timing service, correlation analysis of eLoran timing error, the eLoran differential timing method and verification of eLoran differential timing service) without conclusions are well described and they do not raise any doubts. In terms of the literature review is sufficient for the letter needs (24 positions), all of which are research articles from recognized scientific journals, such as: GPS World, IEEE Antennas and Wireless Propagation Letters, IEEE Transactions on Aerospace and Electronic Systems, IEEE Transactions on Antennas and Propagation, and others. Moreover, I would like to point out that the publications cited are related to the subject of this letter (Loran and signal propagation delay). However, in the paper make the following changes:
- The letter is not at the highest editorial level. It would be necessary to improve, e.g. letter size, quality of the figures, table arrangement and numbering, as well as typos.
- The conclusions should also give information what research has shown (measurement results).
To sum up, after taking into account the above amendments (minor revision), I suppose that this letter is suitable for publication in the Sensors.
Reviewer 4 Report
The paper presents theoritical expressions of propagation delay of eLoran : propagation delay of eLoran signal to the autonomous time service and timing error of eLoran signal, the correlation of eLoran timing service error.
1.The paper is clearly written but there are some text editing mistakes for instance :
* Line 29
* Line 35
* Line 85 and more
Please recheck especially space after a dot at the end of the sentence and also subsection title. Many subsection titles lacks space between the numerical value and the description of the subsection.
2. In section 4, there two Figures with same Figure number (Fig.3) in manuscript. It is a little bit confusing and difficult to follow.
3. I will wish authors to discuss the advantages of eLoran system over GNSS system in Introduction section and some relevant points that make experts to focus much on GNSS eLoran system.
Round 2
Reviewer 1 Report
General remarks:
- The 1st review of abstract and introduction has proposed: “It should also be mentioned that there are different views on eLoran as a GNSS backup system worldwide, including the maritime community. This should be corrected in the abstract and in the introduction of the paper.” In the 2nd version a partly revision of abstract and introduction has been done. However still statements are contained, which one cannot agree, e.g. : “In most communities around the world, eLoran is selected as the most promising backup for GNSS in case the use of navigation satellite signals is denied.” or “The differential method is applied to the eLoran
system called eLoran differential timing technology, which has been widely used in maritime affairs of eLoran.” - The 1st review proposed to check spelling and grammar throughout the paper and to complete sentences. Although it has been said that the text has been checked by a native English speaker, it must be said that here too, a significant improvement is needed. This also applies to the choice of technical terms. For example:31-32: “That is the reason why the eLoran is used as the supplement system of GNSS, Unfortunately, eLoran is less accurate than GNSS”
- …..
- 16-18: “Then the eLoran differential method is propose, and in the difference station the theoretical calculation is combined with the measurment of the signal delay to obtain the difference information, which is sent to the users to adjust the prediction delay and promote the eLoran timing precision”
- In general, the formulas are now better explained in terms of the formula symbols used and the meaning. However, the need is seen to work on the formatting of the font size (text, numbers, equation). It is also seen as necessary that beyond the individual sections a consistent use of technical formulations and formula symbols has to be ensured.
Further remarks:
- Chapter 2 starts now with a short introduction, what eLoran is and informs about the basic signal structure. Figure 1 might be sufficient for this purpose. Abbreviations like BPL must be explained. The formulas are now better explained regarding used formula signs. However, especially here (text, figures) the formatting of the font size must be worked on.
- It is seen that a revision of section 3.1 has been done to explain more measured values, error sizes and error estimates. Please check the consistent use of formula signs: e.g. “tg is calculated with the formular (1) (2) (3), and the deviation from the true value is about 500ns.”, but formula (1) uses Tp as propagation delay.
- In section 4 the reference station of differential eLoran is called difference station, why? The associated formulas use M and U as index, why M for difference station? In chapter 5 the term reference station is used, why not here?
- In section 4.3 it is proposed to reduce the receiving error by measuring the relative delay of the receivers. Is this applicable in practice? What is the absolute error?
- The title of chapter 4.4 and the content should be harmonized. A spatial separation between tables and formula (13) would improve the readability.
Reviewer 2 Report
I am sorry to say that the language is still far from acceptable level. The poor language results in many ambiguities hence it is difficult to follow the reasoning of the author.
Author Response
- We are very sorry for the spelling and grammar mistakes in this manuscript and inconvenience they caused in your reading. The English editors have been thoroughly checked and revised the manuscript, so we hope it can meet the journal’s standard.
